# On the Generalizability of "Competition of Mechanisms: Tracing How Language Models Handle Facts and Counterfactuals"

**Asen Dotsinski**[*]                                                         *asen.dotsinski@student.uva.nl*
*University of Amsterdam*

**Udit Thakur**[*]                                                              *udit.thakur@student.uva.nl*
*University of Amsterdam*

**Marko Ivanov**                                                             *marko.ivanov@student.uva.nl*
*University of Amsterdam*

**Mohammad Hafeez Khan**                                              *hafeez.khan@student.uva.nl*
*University of Amsterdam*

**Maria Heuss**                                                                    *m.c.heuss@uva.nl*
*University of Amsterdam*

**Reviewed on OpenReview:** *https://openreview.net/forum?id=15keyzQj9h*

## Abstract

We present a reproduction study of "Competition of Mechanisms: Tracing How Language Models Handle Facts and Counterfactuals" (Ortu et al., 2024), which investigates competition of mechanisms in language models between factual recall and counterfactual in-context repetition. Our study successfully reproduces their primary findings regarding the localization of factual and counterfactual information, the dominance of attention blocks in mechanism competition, and the specialization of attention heads in handling competing information. We reproduce their results on both GPT-2 (Radford et al., 2019) and Pythia 6.9B (Biderman et al., 2023). We extend their work in three significant directions. First, we explore the generalizability of these findings to even larger models by replicating the experiments on Llama 3.1 8B (Grattafiori et al., 2024), discovering greatly reduced attention head specialization. Second, we investigate the impact of prompt structure by introducing variations where we avoid repeating the counterfactual statement verbatim or we change the premise word, observing a marked decrease in the logit for the counterfactual token. Finally, we test the validity of the authors' claims for prompts of specific domains, discovering that certain categories of prompts skew the results by providing the factual prediction token as part of the subject of the sentence. Overall, we find that the attention head ablation proposed in Ortu et al. (2024) is ineffective for domains that are underrepresented in their dataset, and that the effectiveness varies based on model architecture, prompt structure, domain and task.

## 1 Introduction

Large language models (LLMs) have been growing in size and capabilities over the last few years, showing ever-increasing accuracy on a wide range of downstream tasks (Brown et al., 2020; Liu et al., 2024). Despite

---

[*]Equal contribution

their remarkable achievements, these models remain largely opaque. Gaining deeper insight into the inner workings of LLMs could bring us closer to solving many currently open problems, such as generating faithful explanations for their outputs (Turpin et al., 2023) or accurately evaluating their true capabilities (Shevlane et al., 2023; van der Weij et al., 2024).

Interpretability research has made incremental progress in understanding individual mechanisms within LLMs. Previous studies have investigated specific computational components, such as copy mechanisms (Elhage et al., 2021), factual knowledge recall (Geva et al., 2021; Meng et al., 2022), and attention heads specializing in certain tasks (Voita et al., 2019). However, a critical gap remains in understanding how these diverse mechanisms interact and compete during model inference.

The original work by Ortu et al. (2024) addresses this challenge by using the TransformerLens (Nanda & Bloom, 2022) for analyzing mechanism competition in language models. Their research focuses on the interplay between factual knowledge recall and counterfactual comprehension, discovering a small number of attention heads responsible for the result of the competition.

Our reproduction study aims to achieve two primary objectives. First, we seek to independently verify the original study's findings regarding mechanism competition and information flow patterns. Second, we aim to extend the research by investigating the robustness of the setup in multiple ways. Our extensions include testing the findings on larger model architectures through experiments with Llama 3.1 8B (Grattafiori et al., 2024), modifying the prompt structure to avoid biasing the model by repeating the entire prompt in the same way, and analysing the impact of specific types of prompts based on their subject matter.

## 2 Scope of Reproducibility

Ortu et al. (2024) employ two primary analytical methods: logit inspection using unembedding matrices (nostalgebraist, 2020) and scaling the output of attention heads to measure their importance (similar to attention head patching and knockout (Wang et al.; Geva et al., 2023)). They validate their findings across multiple model scales, focusing predominantly on GPT-2 small (Radford et al., 2019) and Pythia 6.9B (Biderman et al., 2023).

Our contribution lies in the implementation, verification, and extension of the findings of Ortu et al. (2024). We proceed by verifying the key claims made by the authors, which we summarize and re-state as follows:

• **Positional Information Encoding**: Factual and counterfactual information are encoded differently, with factual attributes appearing in subject positions while counterfactual information is encoded in attribute positions. This pattern becomes monotonically stronger with each layer, and particularly so after about half of the model layers have been applied.

• **Attention Block Dominance**: Attention blocks contribute more significantly to mechanism competition than MLP blocks, being primarily responsible for writing both factual and counterfactual information to the final position.

• **Attention Head Specialization**: A few highly activated attention heads are effectively responsible for the outcome of the mechanism competition. They attend to the attribute position regardless of which mechanism they support, and they largely contribute to the competition by promoting or suppressing the counterfactual (i.e. copy) mechanism, leaving the factual (i.e. recall) mechanism largely unchanged.

• **Cross-Model Generalization**: These competition patterns persist across model scales, from GPT-2 small to Pythia 6.9B.

Beyond verifying these core claims, we extend the work through additional investigations:

1. **Model Scaling Study** We evaluate the above claims for a larger and more modern model through experiments with Llama 3.1 8B (Grattafiori et al., 2024).

2. **Prompt Structure Sensitivity** We investigate how prompt structure influences mechanism competition, particularly examining how changing the structure of the prediction sentence from the counterfactual one impacts the model's propensity to repeat the counterfactual claim.

3. **Premise Word Sensitivity** We investigate the effect of different "premise words" (i.e. changing the "Redefine:" at the start of the prompt with other words) on the propensity of the model to predict the factual versus the counterfactual token. We also confirm whether this holds up after ablating the attention heads that are evaluated by Ortu et al. (2024) to play the largest role in this ratio.

4. **Domain Robustness** We analyze the effect of the prompt domain on the findings of the original study. A robust experiment setup should be invariant to the specific topic of the prompts.

## 3 Methodology

While the code from the original study is available, it is distributed across multiple branches for different experiments, with many scripts requiring significant modifications to function properly. We consolidate and improve the codebase, converting plotting scripts from R to Python and rewriting visualization code to ensure reproducibility. [1]

### 3.1 Model Implementation

Our reproduction study focuses on two primary models used in the original paper: GPT-2 small and Pythia 6.9B. To test the generalizability of the results, we also run the same experiments on Llama 3.1 8B. Following Ortu et al. (2024), we use these models as provided by the TransformerLens (Nanda & Bloom, 2022) library for mechanistic interpretation. The library provides support for analyzing the behaviour of specific logit outputs, preserving the original pretrained model weights as found in the HuggingFace Transformers (Wolf et al., 2020) library, without any fine-tuning.

### 3.2 Datasets and Processing

#### 3.2.1 "Redefine" datasets

The dataset used in Ortu et al. (2024) is an augmented version of the COUNTERFACT (Meng et al., 2022) dataset. COUNTERFACT contains almost 22,000 prompts, together with their factually correct continuation, as well as a plausible, but factually incorrect one. An example prompt is "Wallmerod, a town in", with the factual continuation being "Germany" and the counterfactual one being "Belgium".

Ortu et al. (2024) derive two datasets from COUNTERFACT with approximately 10,000 generation prompts each, one for GPT-2 small and one for Pythia 6.9B. The authors state the data is filtered per model so "the attributes are represented by a single token and the model completes the sentence in a factually accurate manner" (the attribute referring to the prompt continuation). For example, the prompt "iPhone is developed by" is included in the GPT-2 small dataset, because GPT-2 small assigns the highest probability to the the token corresponding to "Apple", which is the correct company, and "Apple" fits inside a single token.

Ortu et al. (2024) further modify the prompts they use to fit the needs of their study. They ensure that each prompt expresses a relation $r$ between a subject $s$ and an attribute $a$ in the format $(s, r, a)$. The example iPhone prompt fits this criteria, since $s$ = "iPhone", $r$ = "was developed by", and $a$ = "Apple".

As discussed, for each $(s, r)$ instance, the dataset provides two values for the attribute $a$: a factual token $t_{\text{fact}}$ representing the true attribute, and a counterfactual token $t_{\text{cofa}}$ representing an alternative, false attribute. Under this notation, the final prompt structure used by Ortu et al. (2024) for all experiments is: "Redefine: {s} {r} {tcofa}. {s} {r}"

---

[1] The code for our reproduction study and extensions is available at https://github.com/asendotsinski/comp-mech-generalizability.

For example: `"Redefine: iPhone was developed by Google. iPhone was developed by"` , where the model predicts the next token as either `"Apple"` or `"Google"`.

The authors claim multiple times in Ortu et al. (2024) that the data was filtered so the models correctly predict the factual token for the base prompt (without adding the "Redefine:..." part). However, we verified that 9,137 of the 10,000 prompts in the dataset for Pythia 6.9B and only 5,179 of the 10,000 prompts in the dataset for GPT-2 small have the factual token as the top prediction for their respective model.

We take several steps to remedy this. First, since the prompts from the two datasets do not fully overlap, we combine them into a single prompt bank. Then, we create a unique dataset for each model we test (GPT-2 small, Pythia 6.9B and Llama 3.1 8B), filtering the prompts from the prompt bank. We ensure that the respective model can predict the factual tokens for the base prompts correctly, and that after adding the "Redefine:..." part, the respective model predicts either the factual or the counterfactual token. This results in a dataset of 6,098 prompts for GPT-2 small, 10,697 for Pythia 6.9B and 13,120 for Llama 3.1 8B. We opted against generating these datasets from COUNTERFACT directly because we wanted our prompts to be as close as possible to those from Ortu et al. (2024).

### 3.2.2 "QnA" datasets

For our extended study, we modify the prompt structure to a direct question format:
`"Redefine: {s} {r} {tcofa}. {interrogative} {rr} {s}? Answer:"` where "interrogative" is a phrase that asks about the attribute (e.g. Who/What/Where) and "rr" is a reformulation of the relation.

For example:
`"Redefine: iPhone was developed by Google. What company developed iPhone? Answer:"`

To generate the reformulation of the prompt into a question, we utilize a version of Google's T5 model, fine-tuned for question generation by Romero (2021). The data for this prompt transformation comes predominantly from the "Redefine" datasets we produce, as discussed in Section 3.2.1. As such, we know that each model answers factually to the base prompts of its respective dataset. For the "QnA" structure we still produce one dataset per model. Each dataset is further filtered so that the reformulated questions are still factually answered by the intended model (i.e., when the "Redefine..." part is not present). Furthermore, we removed any full prompts ("Redefine..." + the question) for which the respective model predicts something different from the factual or the counterfactual token.

To ensure we don't make the datasets too small with our further filtering, we also add additional prompts from a modified version of COUNTERFACT that includes a tag for the subject token (Nanda). The full process yields a dataset of 4329 prompts for GPT-2 small, 8,145 prompts for Pythia 6.9B and 9,447 prompts for Llama 3.1 8B.

A key motivation for this dataset is to test the robustness of the findings when the models are presented with a different prompt structure. In particular, we hypothesize that the "counterfactual" mechanism does not rely on the model reading the word "Redefine" and understanding that it is prompted to produce a counterfactual statement. Rather, it seems intuitive that the models are simply finishing the repeated prompt sentence. By restructuring the second sentence into an interrogative form, then, we expect that the models will be less inclined to predict the counterfactual token, since copying from the first sentence will not be as trivial.

### 3.2.3 "Premises" datasets

Ortu et al. (2024) choose to put the "Redefine" keyword at the start of their prompt without too much justification. To validate this choice, we take our "Redefine" dataset for GPT-2 small (Section 3.2.1) and we substitute the keyword for several others: "Fact Check", "Validate", "Verify", "Review" and "Assess". We then measure what percentage of the prompts get a counterfactual token versus a factual one, also after ablating the attention heads suggested by Ortu et al. (2024) (see Section 3.3.2).

Our hypothesis is that, under the counterfactual mechanism, the model simply sees the same sentence repeated twice and finishes it off the same way, so the start shouldn't affect the rate of prompts finished

in a factual versus counterfactual way. Furthermore, if the counterfactual mechanism doesn't meaningfully change, we should expect the same ablation of attention heads to be just as effective.

### 3.2.4 Domain Analysis

We also analyze the effects of the prompt domain (i.e. category) on the competition of mechanisms in GPT-2 small. To that end, we take the original 10,000-prompt large dataset for GPT-2 and we separate it into 26 different categories like "Autos and Vehicles", "Finance", "News" and others. We classify the prompts using Nvidia's NeMo Curator domain classifier (Nvidia), a model that is based on DeBERTaV3 (He et al.) and trained specifically for this purpose. The dataset turns out to be rather skewed, with just two categories - "Autos and Vehicles" and "Computers and Electronics" - accounting for over half of all prompts, while three categories - "Adult", "Beauty and Fitness" and "Pets and Animals" - having no entries.

To extend this analysis beyond COUNTERFACT, we also conduct it on the MQuAKE dataset (Zhong et al., 2023). It has a similar prompt structure to COUNTERFACT, but its predominant categories as predicted by NeMo Curator are very different, covering mostly "Arts and Entertainment", "Sports" and "Law and Government". We merged all dataset variants, retained only the "requested rewrite" entries, and removed duplicates. While we were able to extract 8,346 samples from MQuAKE, our filtering substantially reduced that number. GPT-2 small was able to correctly predict only about 500 of the base prompts. After removing prompts where the model did not predict the factual or the counterfactual token for the "Redefine" prompt, we were left with just 228 prompts. We decided to still include these results, recognizing that they are tentative due to the low sample size.

The goal of these datasets is to allow us to compare the effect of the originally proposed ablation of attention heads, as discussed in Section 3.3.2, to data of different domains. Such analysis would test the generalizability of the original findings; if the proposed ablation of attention heads works only for the most prominent categories of the original dataset, this would imply that the most impactful attention heads vary with the content of the input data. Practically, this would make the ablation of attention heads conditional on a very tight control of the topic of the input prompts.

## 3.3 Experimental Setup

All of the experiments in Ortu et al. (2024) are replicated using (a version of) the code supplementing the original paper. We take the liberty of adding missing experiments from different branches of the repository, as well as fixing coding errors that lead to result mismatches. We replicate the experiments outlined in Sections 6.1-6.4 of Ortu et al. (2024), as we feel they are the most central to the presented claims (Section 2).

Following the authors' methodology, we use two primary evaluation methods to analyze mechanism competition: logit inspection and attention modification. We use both for all of our experiments, except for the "Premises" datasets (Section 3.2.3), where we conduct only the ablation modification. In the interest of time, we perform the "Premises" and domain analysis only on GPT-2 small.

### 3.3.1 Logit Inspection

As in the original study, we use the TransformerLens (Nanda & Bloom, 2022) library for logit inspection, which implements the logit lens method (nostalgebraist, 2020). Here we present the intuition behind the method.

Modern LLMs rely on a residual stream $x$. The residual stream is incrementally updated by each of $L$ attention blocks $a^l$ and MLPs $m^l$:

$$x_i^l = x_i^{l-1} + a_i^l + m_i^l \tag{1}$$

where $x_i^l$ is the residual stream for the token at position $i$ after the $l$-th layer of the LLM. To obtain the final prediction, the residual stream must be multiplied by an unembedding matrix $W_U$, to bring its dimension back to the size of the vocabulary:

$$\tilde{x}_i^L = x_i^L \cdot W_U \tag{2}$$

The intuition behind the logit lens method is that the residual stream gradually converges to the final distribution over the layers of the network, often approximating it many layers before the end. Thus we apply the unembedding matrix at earlier layers and inspect the results:

$$\tilde{x}_i^l = x_i^l \cdot W_U \tag{3}$$

where $\tilde{x}_i^l$ represents the predicted token at position $i$ in the residual stream. This projection allows us to track the logits of the factual token $t_{\text{fact}}$ and counterfactual token $t_{\text{cofa}}$ at different points in the network. Furthermore, the TransformerLens library enables attribution of these values to specific attention heads, positions, and MLP blocks, providing fine-grained analysis of how each component influences the final prediction.

### 3.3.2 Attention Modification

As in the original methodology, attention heads that have a high contribution to the mechanistic competition can be modified to confirm the effects on the final prediction. For attention head $h$ in layer $l$, we can scale the weight with which the token in position $i$ attends to the token in position $j$ using:

$$A_{ij}^{hl} \leftarrow \alpha \cdot A_{ij}^{hl}, \text{ where } j < i \tag{4}$$

where $\alpha \in \mathbb{R}$ is a scaling factor.

Deciding which heads to be ablated, how many and with what scaling factor is somewhat arbitrarily decided in Ortu et al. (2024). It is largely based on a qualitative analysis of the attention head attribution scores (as mentioned in Section 3.3.1) for the factual and counterfactual logits predicted at the last position. The heads that increase the counterfactual logit compared to the factual the most are chosen for ablation. To maintain consistency with Ortu et al. (2024), we ablate the attention of the last token to the attribute position in L10H7 and L11H10 for GPT-2 small, and L17H28, L20H18, and L21H8 in Pythia 6.9B, using $\alpha = 5$.

Using TransformerLens on Llama 3.1 8B, as we discuss in the results (Section 4.2), we find a single head that heavily contributes to the counterfactual token (L27H20), and a more spread out factual contribution over multiple heads. We experiment with turning off the counterfactual head entirely ($\alpha = 0$), as well as modifying the two most attention heads that contribute the most to the factual token (L28H15 and L31H14), first using $\alpha = 5$ and then $\alpha = 50$.

### 3.4 Computational Requirements

All of the experiments were performed on the Snellius supercomputer cluster. The exact hardware used is a single NVIDIA H100 (SXM5) GPU coupled with 8 cores of an AMD EPYC 9334 CPU. The running time of the experimentation depends on the size of the model. In total, to achieve the final results for all models, 57 GPU and CPU hours were used, with an additional 40.5 hours used up for testing and failed runs. We also note that the performance of the experiments is mostly CPU bound in terms of computation, so there is a potential improvement if the workload is either offloaded to a GPU or adjusted to allow parallel execution on separate CPU cores.

To estimate the carbon footprint, we can use the average Power Unit Efficiency ($PUE$) for the Netherlands in 2024 of 1.3 according to the State of the Dutch Data Centers Report 2024 by Dutch Data Center Association and a Carbon Intensity ($CI$) 0.268kg/kWh for the Dutch power grid according to Our World in Data. We use the SURF User Knowledge Base statistics data to get exact number for the energy consumption $E = 52.80$kWh and a total runtime of 129:01 hours to obtain $CO_2e = CI * PUE * E = 18.39kg$. This is roughly equivalent to driving a typical passenger car for 130 km.

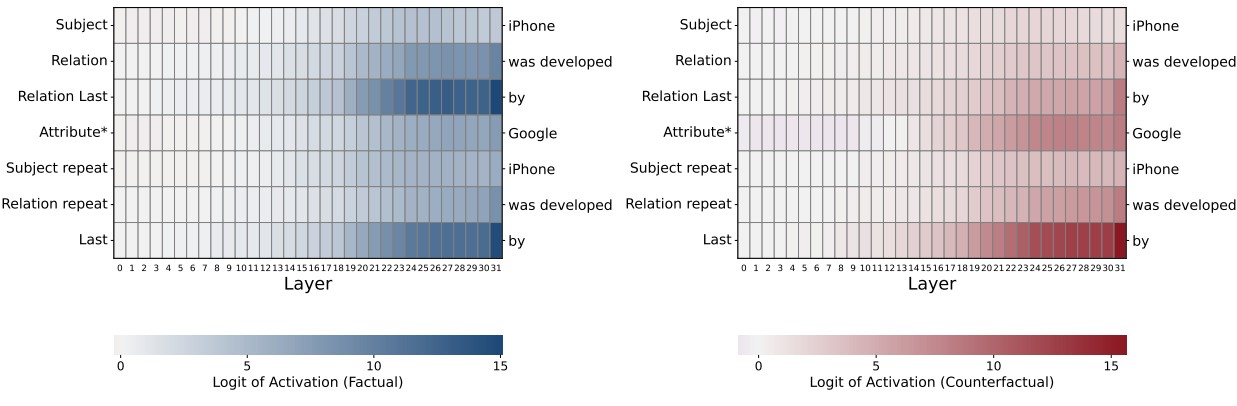

(a) The logit values for the factual token.

(b) The logit values for the counterfactual token.

Figure 1: The logit values for Pythia 6.9B across different positions and layers on the filtered "Redefine" dataset. The activations in the subject positions are notably lower than the other plotted positions.

## 4 Results

### 4.1 Reproduction of Original Claims

Despite the initial discrepancies in the datasets provided by Ortu et al. (2024), as discussed in Section 3.2.1, after proper filtering, we successfully reproduce the three key mechanistic findings from the original paper, both for GPT-2 small and Pythia 6.9B.

**Positional Information Encoding:** Our results confirm that, in the early layers of GPT-2 small, factual information is spread out but most strongly encoded in the subject position, while counterfactual information is predominantly encoded in the attribute position (Figure 6 in the Appendix). As in the original study, in later layers both types of information become strongly concentrated in the last token position, with logit values showing a monotonic increase across layers.

We also replicate the results for Pythia 6.9B, with no significant deviations. The early layers (up to about layer 17) have a relatively small impact on the factual or counterfactual logit when inspected, compared to the layers after that (Figure 1). However, the claim that factual information is stored in the subject position does not seem to be backed up by our results or those of the original paper, regardless of the inspected layer. Instead, the positions most strongly influencing the factual prediction seem to be the last token of the first relation and the tokens of the first relation themselves.

**Attention Block Dominance**: As in Ortu et al. (2024), we inspect the margin of the added logit of $t_{\mathrm{cofa}}$ over that of $t_{\mathrm{fact}}$ in each block with regards to the final token prediction, represented by

$$\Delta_{\mathrm{cofa}} := \mathrm{BlockLogit}(t_{\mathrm{cofa}}) - \mathrm{BlockLogit}(t_{\mathrm{fact}}).$$

The logit contribution of an attention block is the sum over the contribution of all attention heads. A positive value of $\Delta_{\mathrm{cofa}}$ for a block signals that it favours the counterfactual token in the competition, and a negative value indicates support for the factual token.

Our findings closely match those of Ortu et al. (2024). For GPT-2 small, we observe minimal competition ($\Delta_{\mathrm{cofa}} \approx 0$) in early layers (0-4), followed by pronounced competition in later layers (5-11). Attention blocks show stronger contributions with $\Delta_{\mathrm{cofa}}$ peaks of approximately 1.3 in layers 7 and 9, compared to more modest MLP block contributions with maximum values around 0.7 (Figure 8 in the Appendix). The results for Pythia 6.9B also match the original study, with the attention blocks dominating but with overall smaller $\Delta_{\mathrm{cofa}}$ values spread over a larger number of layers (Figure 11 in the Appendix).

**Attention Head Specialization:** We confirm the claim of attention head specialization for GPT-2 small and Pythia 6.9B. A small number of heads in the latter half of the GPT-2 small network dominate the rest

in deciding the outcome of the mechanistic competition (Figure 7 in the Appendix) . Furthermore, all highly activated heads show strong attention to the attribute position, with the heads that favour the factual outcome achieving this by reducing the value for the counterfactual logit. This validates the claim from Ortu et al. (2024) that the factual token does not get promoted directly, but rather through counterfactual suppression. We replicate similar results for Pythia 6.9B (Figure 10 in the Appendix), with a larger number of specialised heads due to the increased network size.

## 4.2 Competition Mechanism in Llama 3.1 8B

The results for Llama 3.1 8B show a clear competition of the mechanisms, similar to the earlier results for GPT-2 small and Pythia 6.9B. Similarly to Pythia, we see that factual information is stored in the position for the last token of the relation, while the counterfactual information resides in the position of the counterfactual, as can be seen by the positional logit activations on Figure 2. However, there are a few notable differences beyond that.

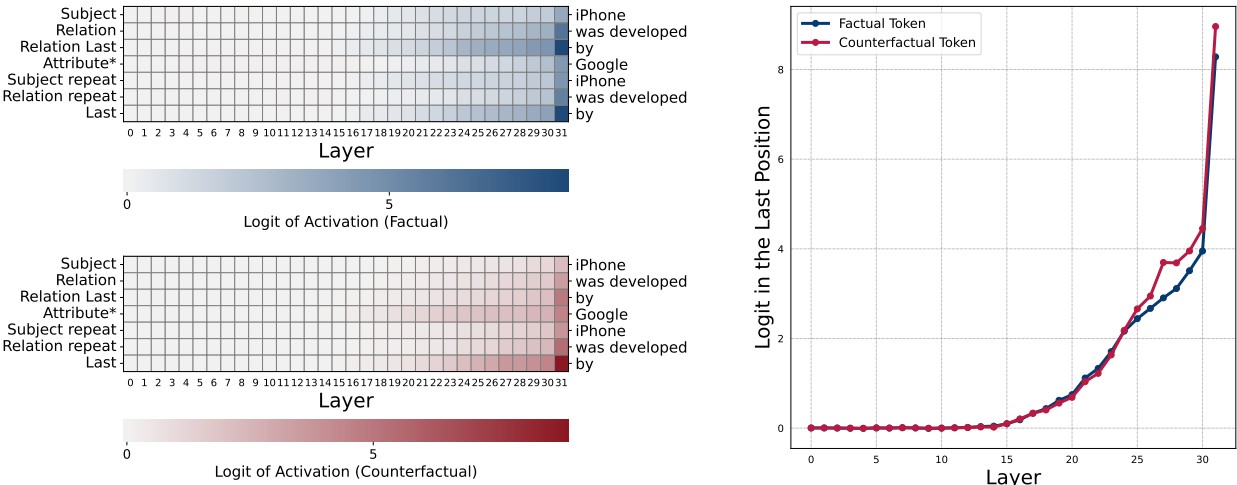

Figure 2: Logit inspection for Llama 3.1 8B on the GPT-2 small "Redefine" dataset.

Llama 3.1 8B is of similar size to Pythia 6.9B and so it was expected that, similarly, the early layers would be less impactful towards the final prediction. However, with Llama we see practically no effect on the factual or counterfactual token for the first 15 layers, followed by a slow rise until layer 20, where the pattern starts to resemble Pythia. This is mirrored in the attention and MLP block attributions, where we see very little to no activity until about layer 23. This lack of early layer activations could be explained by some more recent work suggesting that the logit lens method is unreliable for larger models (Belrose et al., 2023), although the relative difference of effectiveness across Llama and Pythia remains unexpected.

Furthermore, we see a very sharp activation increase in the very last layer of Llama 3.1 8B, accounting for approximately half of the logit values of the factual and counterfactual token predictions. This behaviour is even harder to explain, and could point towards a model-dependent process that is not captured by the current experimental setup.

We find more evidence of such a deficiency when we attempt to ablate the most impactful attention heads. Not only do we see low $\Delta_{\text{cofa}}$ for all but one counterfactual head (Figure 13 in the Appendix), but ablating the model by disabling that head (Figure 16 in the Appendix) or by boosting the two most contributing factual heads with $\alpha = 5$ (Figure 15 in the Appendix) were both only mildly effective, increasing the share of factually predicted prompts from 15% to 22%. Either the logit lens method fails to discover the highly specialized attention heads, or they are simply not as concentrated for Llama 3.1 8B.

### 4.3 Impact of Question-Answering Format

We investigate the robustness of mechanism competition by reformulating the original completion tasks into explicit questions.

Overall, in all tested models (GPT-2 small, Pythia 6.9B and Llama 3.1 8B) we see patterns of competition consistent with our results on the "Redefine" datasets (Appendix B). A noticeable difference is that logit activations for the counterfactual tokens are reduced in GPT-2 (Figure 17 in the Appendix) and Pythia (20 in the Appendix), especially when observing the earlier layers. For Llama, which already had very low activations in earlier layers, there is a smaller difference (Figure 23 in the Appendix).

These findings imply two things. Firstly, the repetitive structure of the original prompt conditions the models to repeat the whole sentence verbatim, supporting the counterfactual mechanism. In GPT-2, this new prompt narrows the gap between the logits so that they are almost equal, while for Pythia the "QnA" prompt gets an on-average higher activation for the factual token, reversing the initial findings.

Secondly, the size and architecture of the model has a significant impact on the perceived difference between the prompt responses. For GPT-2 small, which is a fairly limited model, the prompt structure resulted in a reduction of the logit value for the final prediction by about 20%, while for Pythia 6.9B and Llama 3.1 8B, the reduction was less than 10%. This suggests that larger models are better at extracting the semantic meaning of the prompt (which remains unchanged) and are thus less likely to fall into patterns of blind repetition. Furthermore, Llama seems to be the least affected by the prompt change, which might be explained in part by the low activations in the early layers of the model. This would support the idea that Llama relies on a mechanism our setup is not able to analyze fully in the early layers, one that is more resistant to repetitive prompts.

Finally, to compare the impact of the "QnA" prompt structure more directly with the different premises, we ablate the two most contributing attention heads for GPT-2 small as suggested by Ortu et al. (2024). The results at the bottom of Table 1 show that the effect of the "QnA" prompt structure is significantly more important than the premise word, leading to a factual token prediction in just under half of all prompts. Ablating the attention heads leads to an even more extreme result, with just 175 prompts receiving a counterfactual prediction. However, it must be noted that after the ablation a significant number of prompts (over 1,300) receive a prediction that is neither factual nor counterfactual. This enforces the claim by Ortu et al. (2024) that the "factual" attention heads work by suppressing the copy mechanism, rather than by enforcing factual recall.

### 4.4 Impact of Premise Words

| Premise | Baseline | | | Ablated | | |
|---|---|---|---|---|---|---|
| | #Factual | #Counterfact | %Factual | #Factual | #Counterfact | %Factual |
| Redefine | 304 | 5794 | 4.98% | 2722 | 3177 | 46.15% |
| Assess | 388 | 5688 | 6.38% | 2843 | 2945 | 49.13% |
| Fact Check | 206 | 5861 | 3.39% | 1883 | 3980 | 32.12% |
| Review | 116 | 5986 | **1.90%** | 1824 | 4093 | **30.83%** |
| Validate | 380 | 5676 | 6.28% | 2888 | 2870 | 50.16% |
| Verify | 259 | 5818 | 4.26% | 2487 | 3332 | 42.74% |
| Redefine, QnA prompt structure | 2158 | 2171 | **49.8%** | 2751 | 175 | **94.02%*** |

Table 1: Comparison of factual predictions (and the QnA prompt structure) across different premises in baseline and ablated conditions (GPT-2 small, $\alpha = 5$, L10H7 and L11H10). Note the reduction of total factual or counterfactual predictions for the ablated QnA prompts.

The number of factual versus counterfactual predictions of GPT-2 small on the "Premises" dataset (Section 3.2.3) can be seen in Table 1. Without ablating the attention heads, we see that "Assess" and "Validate" have the highest numbers of factual predictions at about 6.3%, while "Review" has the lowest at just 1.9%. While the counterfactual predictions dominate for all premise words, the range of values suggest that these results are not just due to numerical instabilities. The particular over and underperformers are also not random: "Assess" and "Validate" are words that imply we are unsure of the validity of the following sentence, while "Review" and "Fact Check" are words that are often found at the start of the title of a journalistic article that has found evidence for a particular claim.

Looking at the ablated numbers, the trends remain the same, but the gap widens. "Validate" and "Assess" now produce a factual claim almost half the time, while "Review" does so for only 30.8% of the prompts. This data is also surprising, because it implies that the effectiveness of the ablation depends largely on the premise. It seems plausible that the premise words serve as triggers, encouraging the copy mechanism to varying degrees, and that their effect is amplified by the attention head ablation.

## 4.5 Impact of Domain

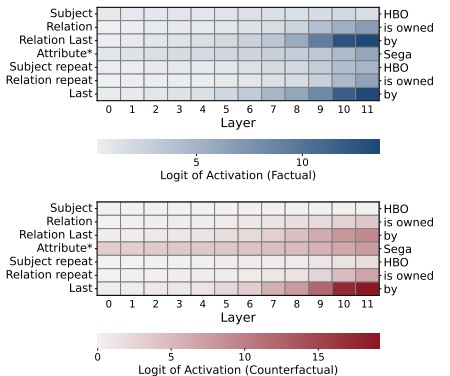
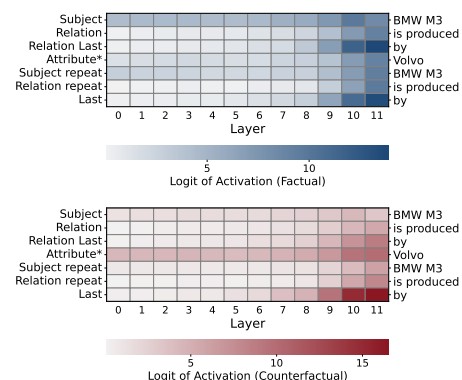

(a) Logits per position for the domain "Arts and Entertainment".

(b) Logits per position for the domain "Autos and Vehicles".

Figure 3: Logit inspection for GPT-2 small on different domains. The results for "Autos and Vehicles" closely match the ones reported in Ortu et al. (2024), while the results for "Arts and Entertainment" show that neither the factual nor the counterfactual logits are strong in the subject position.

Our domain-specific investigation reveals significant variations in mechanism competition across different knowledge domains (Figure 3). The results for overrepresented domains like "Autos and Vehicles" and "Computers and Electronics" match closely with the results of the original study. However, underrepresented domains like "Arts and Entertainment" and "Law and Government" show higher counterfactual token activations throughout the network, and pay very little attention to the first subject token. Furthermore, in these domains the attention heads that most contribute to either the factual or the counterfactual token tend to differ slightly from those in the original study.

Further analysis provides a possible explanation for this clear domain split: practically all prompts from "Autos and Vehicles" and "Computers and Electronics" ask about which company creates a certain product, and these product names predominantly begin with the name of the company (i.e. the factual token) itself. Examples include the "Nintendo DS Lite", "Honda Aviator" and "Microsoft Office 2007". In comparison, the prompts from "Arts and Entertainment" tend to ask about the citizenship of a particular celebrity or the language in which a movie like "Star Wars: Episode III" was originally published, which do not contain the factual token explicitly. This would suggest that the reliance on the subject token for the factual prediction mechanism, as we saw it in our reproduction of the original claims (Section 4.1), is spurious and does not apply to prompts that do not leak the factual token.

To quantify the effects of these findings, we ablated attention heads L10H7 and L11H10 with $\alpha = 5$ when attending to the attribute position for GPT-2 small, as described in Ortu et al. (2024), for every domain. They

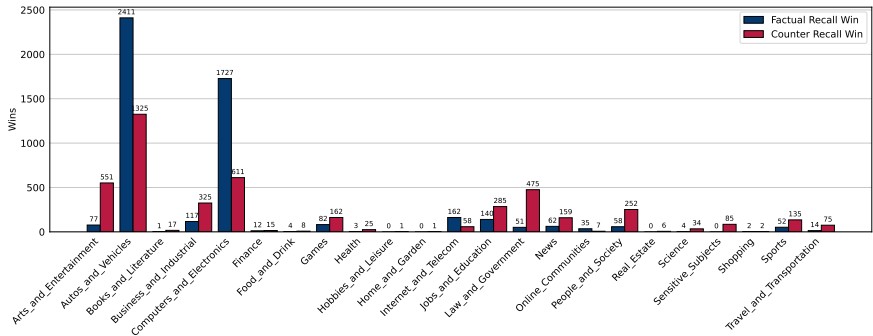

Figure 4: GPT-2 domain-level wins at attribute position after ablation of L10H7 and L11H10

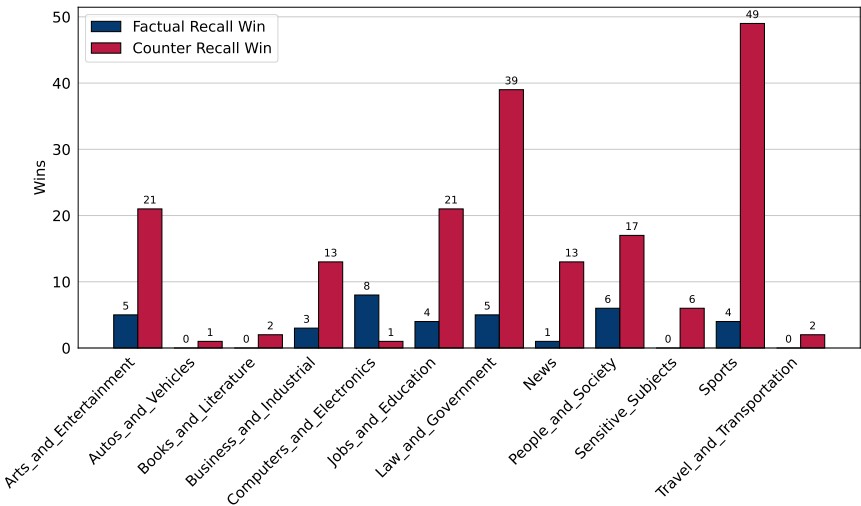

Figure 5: Per domain ablation results for GPT2 on the filtered MQuAKE dataset (GPT-2 small, $\alpha = 5$, L10H7 and L11H10)

found these two attention heads to dominate the contribution towards the counterfactual token predictions, and that ablating them increases factual predictions from 4% to 50%. As can be seen in Figure 4, this ablation leads to the dominance of the factual on the categories "Autos and Vehicles", "Computers and Electronics" and "Internet and Telecom", while in all other categories the counterfactual token is still mostly predicted. These results seem to confirm that prompts from different domains activate different attention heads, making ablation less effective. As such, the identity of these highly specialised attention heads is not just model-dependent, but task- and even domain-dependent.

Ablating the same heads for GPT-2 small on the MQuAKE-CF dataset, we see that the effect is even more subdued (Figure 5). As with the dataset from Ortu et al. (2024), we see the largest relative increase in "Computers and Electronics", going from 2 factual and 7 counterfactual predictions before the ablation to 8 factual and 1 counterfactual prediction after the ablation. Besides "Business and Industrial" and, surprisingly, "Arts and Entertainment", the rest of the categories saw minimal relative increase in factual predictions. In particular, the number of counterfactual predictions for "People and Society" increased after the ablation. While these results are in line with our expectations, the size of the filtered dataset is small. Further work is warranted on the domain specificity of different mechanisms.

# 5 Discussion and Conclusion

## 5.1 Main Findings and Reproduction

Our reproduction study successfully validates the core claims made by Ortu et al. (2024) regarding mechanism competition in language models. We confirm their key findings about information flow patterns, attention block dominance, and specialized attention heads across both GPT-2 small and Pythia 6.9B models. While our analysis reveals some dataset limitations regarding factual token predictions, and the role of the subject token as a storage for factual information is called into question, the main findings generally hold up.

## 5.2 Extended Findings

Our experiment extensions reveal several key insights. Firstly, experiments with Llama 3.1 8B show that while basic competition patterns persist in newer architectures, the dynamics vary significantly, with delayed activation patterns and pronounced final-layer effects.

Secondly, investigation of QnA-format prompts demonstrate that the original prompt structure artificially strengthens the counterfactual (copy) mechanism through context repetition. We observed that this prompt reformulation is roughly as effective as the attention head ablation proposed by Ortu et al. (2024) for GPT-2 small. While we test only a handful of models, it seems that this effect is more pronounced in smaller, less trained ones.

Next, our analysis of different premise words further confirms that the structure of the provided prompt can be significant for influencing models to predict a factual or a counterfactual token. While ablating attention heads is effective for every premise, some premises seem to increase this effect dramatically, suggesting that the context around the prompt works to further encourage or suppress the copy mechanism.

Finally, our domain analysis uncovers that the dataset's bias toward automotive and technology domains heavily influences the results reported in Ortu et al. (2024). The predominant appearance of factual tokens in the subject names of the prompts from these domains seems to explain the inconsistent importance of subject positions across our different experiments. Furthermore, the varying effectiveness across domains of the attention head ablation proposed in Ortu et al. (2024) suggests mechanism competition is more complex and domain-dependent than previously understood.

## 5.3 Practical Considerations

Both Ortu et al. (2024) and our paper explore a focused but somewhat narrow experimental setup designed to clearly show the competition of mechanisms in LLMs. In this section, we will give our opinion on how our results fit into a more practical context.

When prompting an LLM, there is usually little reason to purposefully distort its representation of factual information. However, an attacker might try to "persuade" the LLM to generate a harmful response that it was specifically trained to refuse by modifying its understanding of the prompt or the prior instructions (Chao et al., 2023). Furthermore, a user might inadvertently influence the model by introducing their own biases and arguments into the prompt.

If one takes our experiments as a proxy for this phenomenon, our results would suggest that smaller LLMs, are rather "impressionable" and readily change their output based on the context, especially when they get to repeat it verbatim. Notably, we saw that premise words like "Fact Check:" and "Review:" are particularly potent at getting GPT-2 small to repeat the incorrect information, even though they provide a rather neutral context. However, larger and newer models seem to be more resilient to outright repeating given claims from the prompt. Furthermore, we used base language models for our study, while user-facing LLMs typically undergo additional fine tuning and reinforcement learning that teach them to differentiate between the (biased) user prompts and their own autoregressive output. As such, we are wary of extending our conclusions to fine-tuned models. Future work could try to address this gap.

Another potential use for the competition of mechanisms framework is for controlling the reliance of a model on its provided context. If one can identify a small number of attention heads that are most responsible for the factual outcome of the competition and boosts their output, this could produce models that are less susceptible to adversarial influences from the context (e.g. jailbreaking) and more factually grounded. On the other hand, promoting the copy mechanism could result in a model that is better suited for retrieval-augmented generation (RAG), since the focus there is on using given source texts and not on prior beliefs.

Unfortunately, our work shows there are multiple obstacles towards this goal. First, we note that we were unable to find these few highly specialized attention heads in Llama 3.1 8B using the methodology proposed in Ortu et al. (2024), and we would expect the same to hold for newer and larger LLMs. Second, our experiments with the different prompt structure suggest that careful prompt tuning could be just as, if not more, effective than attention head ablation. While using the two techniques together yields even better results, we could imagine that an adversarial prompt could also severely mitigate the desired effect. Finally, we show that, even when these specialized attention heads can be found, their identity shifts depending on the domain of the prompt. This means that attention head ablation might be viable only for language models trained for very narrow tasks, as opposed to the more generalist LLMs that have become commonplace over the last few years.

### 5.4 Limitations and Future Work

While we validate the fundamental insights from Ortu et al. (2024), our extensions reveal crucial nuances in how these mechanisms depend on domain, architecture, and prompt structure. The key takeaway is that experimental findings for mechanistic interpretability cannot be easily assumed to generalize beyond the specific task and dataset of the setup.

While our findings are descriptive, this work leaves a few open questions for future research. Firstly, it is not clear whether larger models like Llama 3.1 8B distribute their work more evenly through early attention heads, leading to less specialization, or whether our logit lens analysis is simply insufficient to capture some other emerging pattern. Newer interpretability techniques, such as circuit analysis (Conmy et al., 2023) or the use of a tuned lens (Belrose et al., 2023), could shine light on the matter.

Secondly, while our work suggests that particular dataset domains have a large impact on which attention heads get activated, our analysis is still rather cursory. It remains to be seen to what degree the source of the discrepancy lies within the subject containing the factual token, the shift of topic words in the prompts, or within the task of the prompt itself, which also varies with the domain. It is possible that prompts of some domains are worded in a way that influences the predicted next token, or they might just be intrinsically harder to answer factually due to being more niche in topic.

Finally, a lot of our extended findings rely exclusively on results from GPT-2 small on a dataset derived from Ortu et al. (2024), and our filtered version of the MQuAKE-CF dataset (Zhong et al., 2023) is too small for definitive conclusions. Thus, we believe that extending our findings to different, larger datasets and models, as well as new prompt structures and premise words, would be a worthwhile future contribution.

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

# A "Redefine" Dataset Plots

## A.1 GPT-2 small Plots

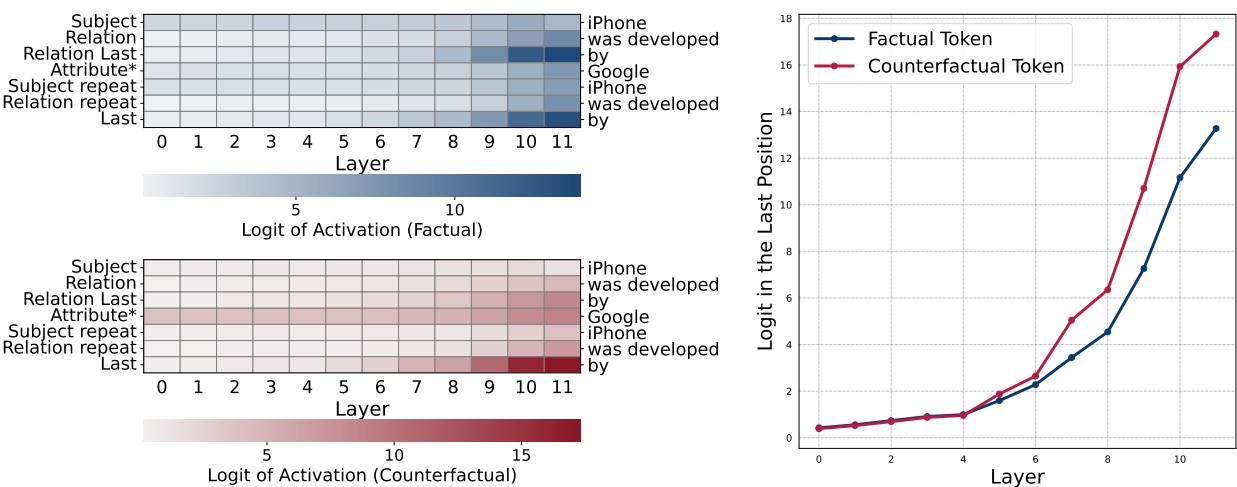

Figure 6: Logit inspection for GPT-2 small on the "Redefine" Dataset.

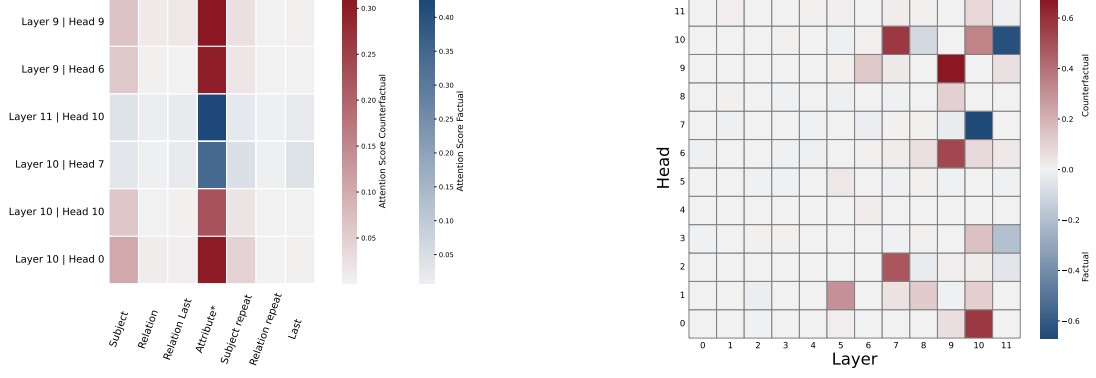

(a) Attention Scores of Key Heads at Last Position.

(b) Contribution to Δcofa by GPT-2 small heads

Figure 7: Logit inspection for GPT-2 small, per attention head, on the "Redefine" Dataset.

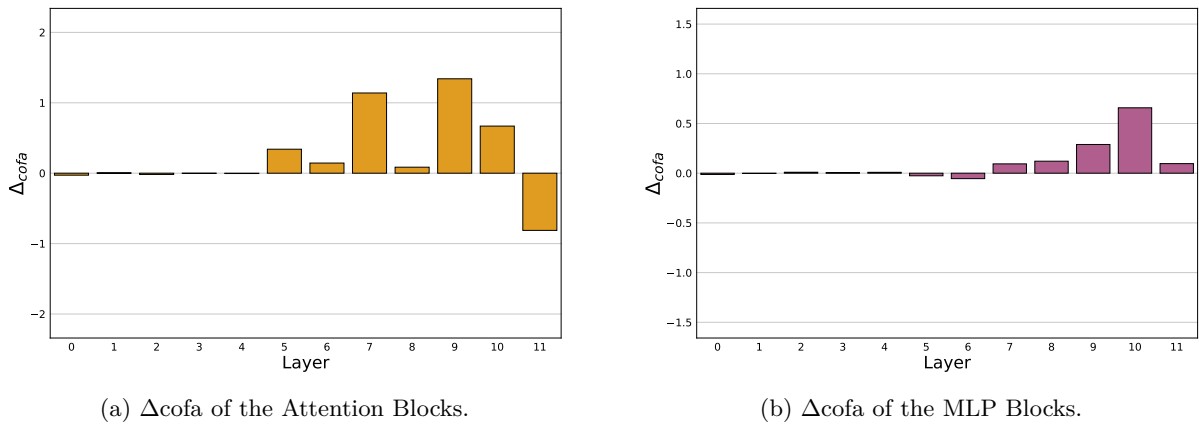

(a) Δcofa of the Attention Blocks.

(b) Δcofa of the MLP Blocks.

Figure 8: Logit inspection of the aggregate impact of attention and MLP blocks on Δcofa for GPT-2 small on the "Redefine" Dataset.

## A.2 Pythia 6.9B Plots

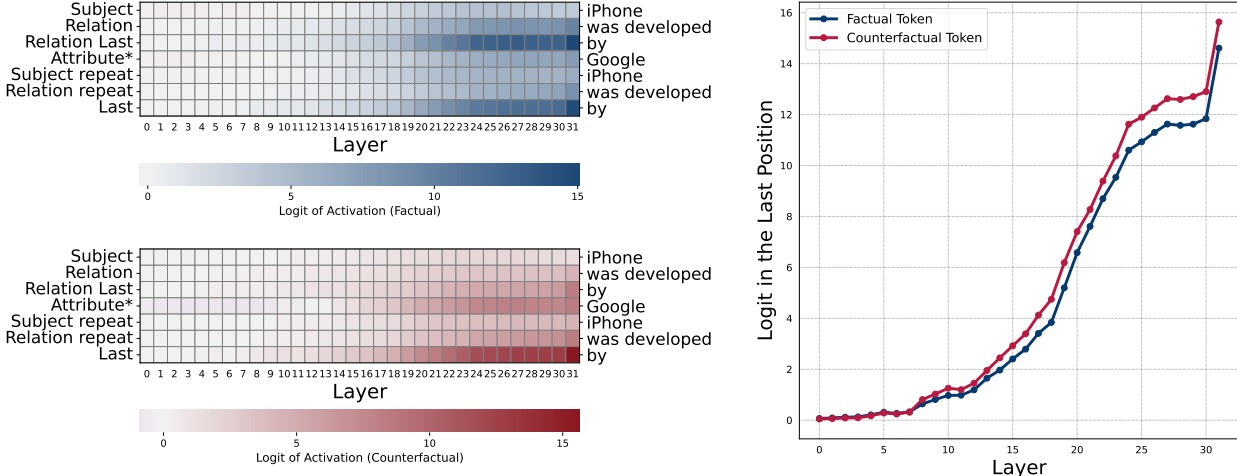

Figure 9: Logit inspection for Pythia 6.9B on the "Redefine" Dataset.

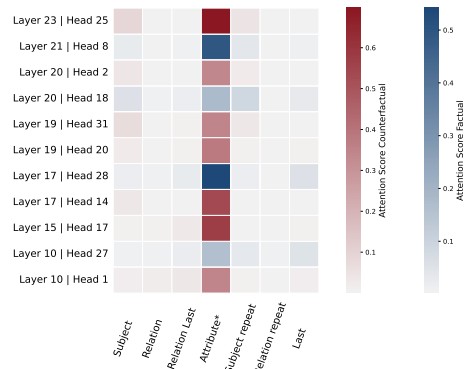

(a) Attention Scores of Key Heads at Last Position

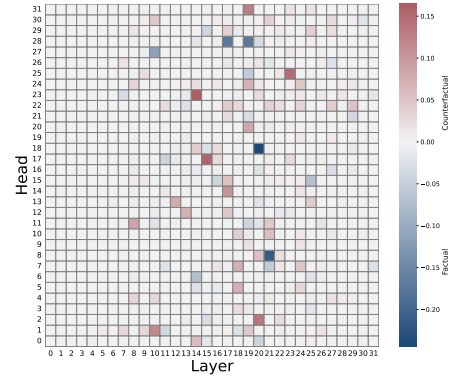

(b) Contribution to Δcofa by Pythia 6.9B Heads

Figure 10: Logit inspection for Pythia 6.9B, per attention head, on the "Redefine" Dataset.

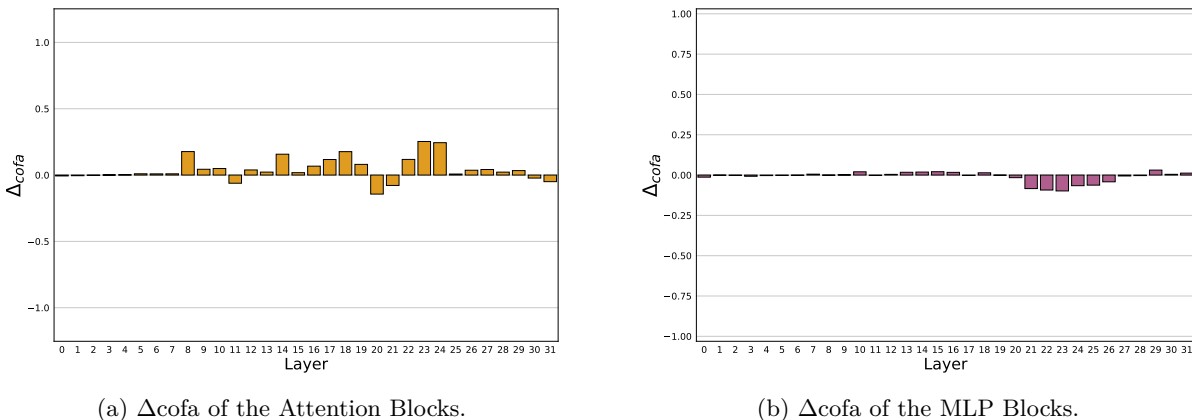

(a) Δcofa of the Attention Blocks.

(b) Δcofa of the MLP Blocks.

Figure 11: Logit inspection of the aggregate impact of attention and MLP blocks on Δcofa for Pythia 6.9B on the "Redefine" Dataset.

## A.3  Llama 3.1 8B Plots

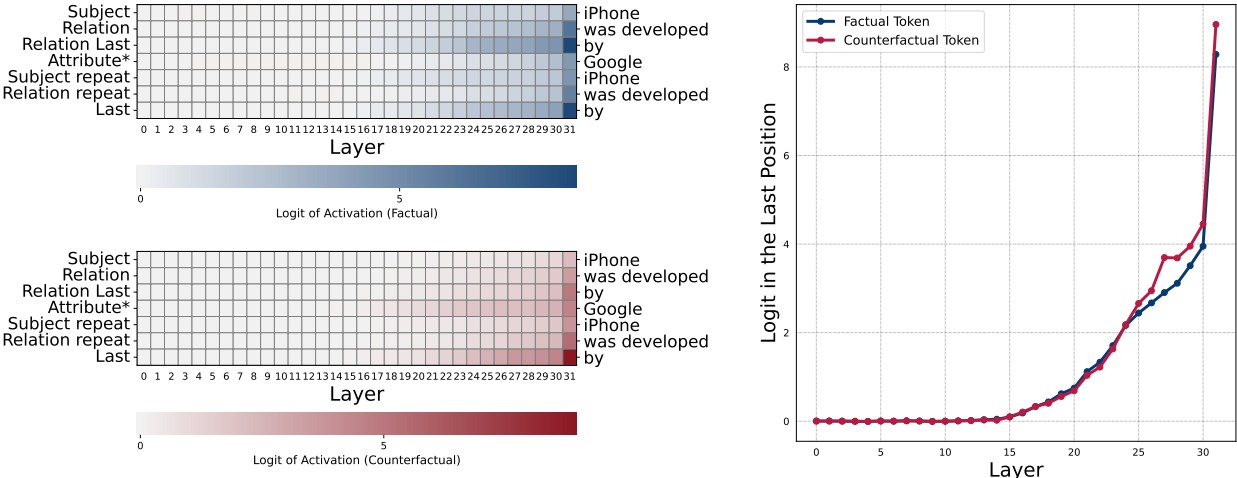

Figure 12: Logit inspection for Llama 3.1 8B on the "Redefine" Dataset.

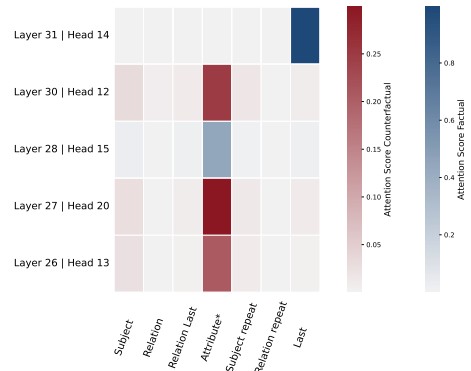

(a) Attention Scores of Key Heads at Last Position

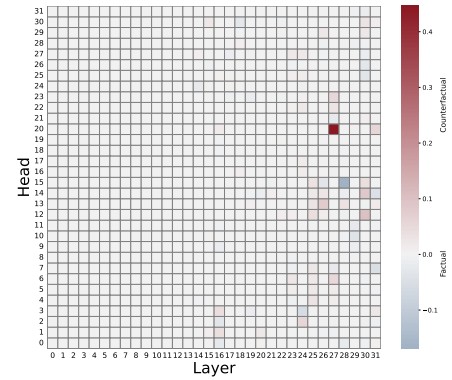

(b) Contribution to Δcofa by Llama 3.1 8B Heads

Figure 13: Logit inspection for Llama 3.1 8B, per attention head, on the "Redefine" Dataset.

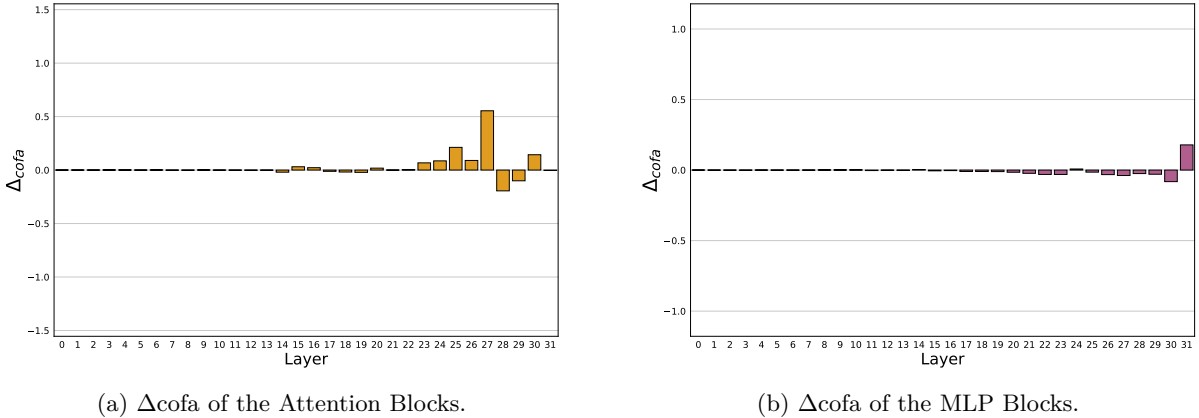

(a) Δcofa of the Attention Blocks.

(b) Δcofa of the MLP Blocks.

Figure 14: Logit inspection of the aggregate impact of attention and MLP blocks on Δcofa for Llama 3.1 8B on the "Redefine" Dataset.

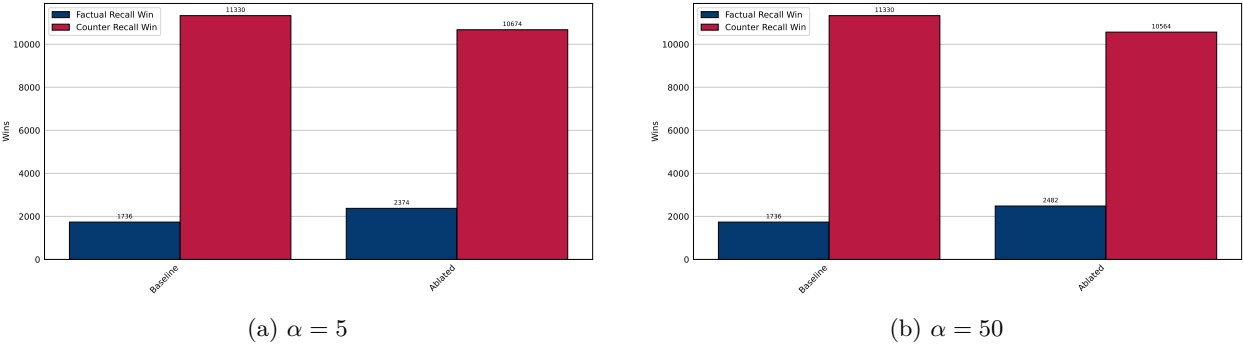

(a) $\alpha = 5$

(b) $\alpha = 50$

Figure 15: Comparison of ablation results (L28H15 and L31H14) for Llama 3.1 8B on the "Redefine" Dataset.

# B QnA Dataset Plots

## B.1 GPT-2 small Plots

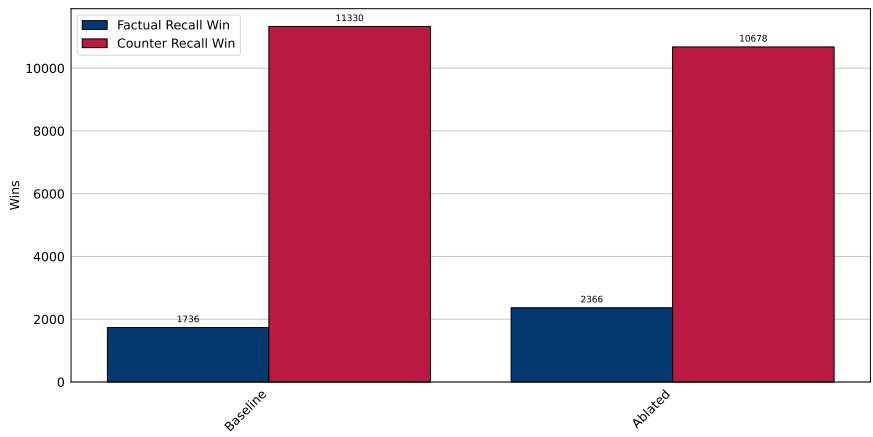

Figure 16: Ablation results (L27H20, $\alpha = 0$) for Llama 3.1 8B on the "Redefine" Dataset.

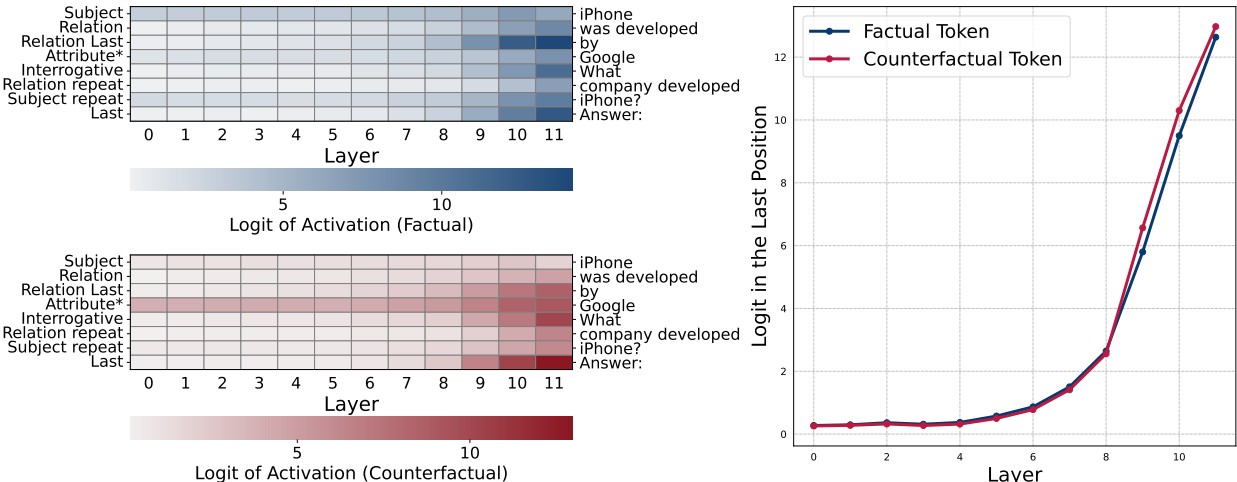

Figure 17: Logit inspection for Gpt2 on the QnA Dataset.

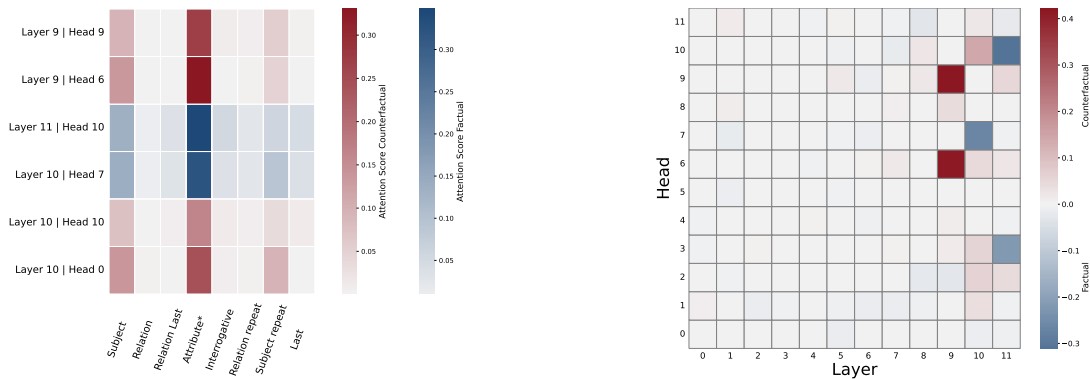

(a) Attention Scores of Key Heads at Last Position for GPT2-small

(b) Contribution to $\Delta$cofa by GPT-2 Heads

Figure 18: Logit inspection for GPT-2 small, per attention head, on the "QnA" Dataset.

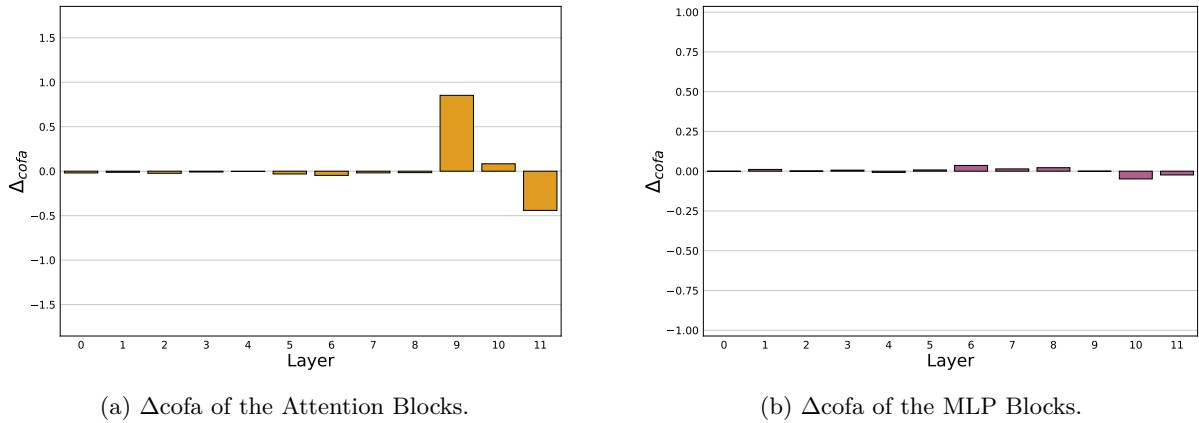

(a) Δcofa of the Attention Blocks.

(b) Δcofa of the MLP Blocks.

Figure 19: Logit inspection of the aggregate impact of attention and MLP blocks on Δcofa for GPT-2 small on the "QnA" Dataset.

## B.2 Pythia 6.9B Plots

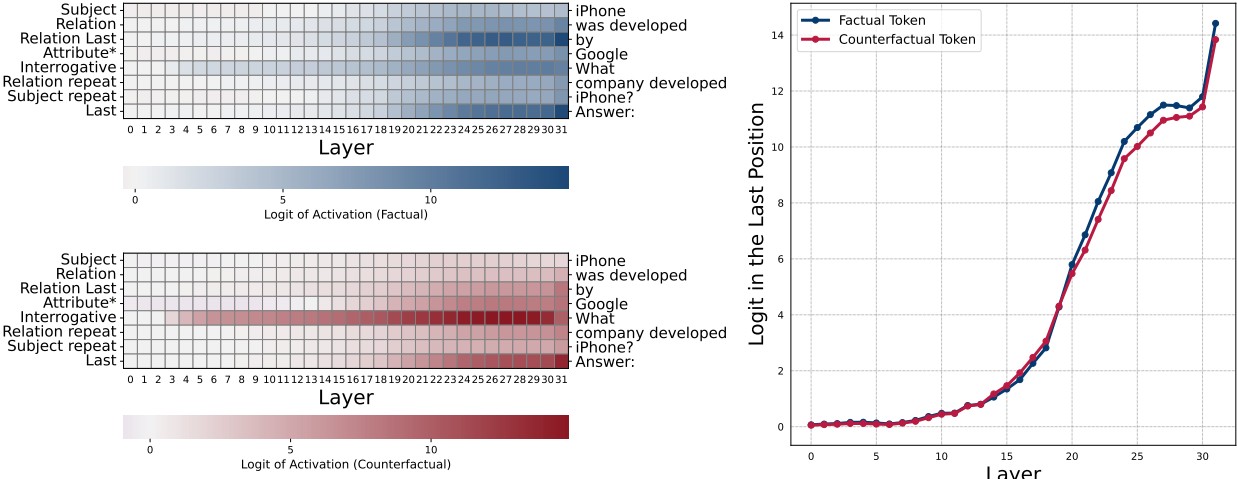

Figure 20: Logit inspection for Pythia 6.9B on the "QnA" Dataset.

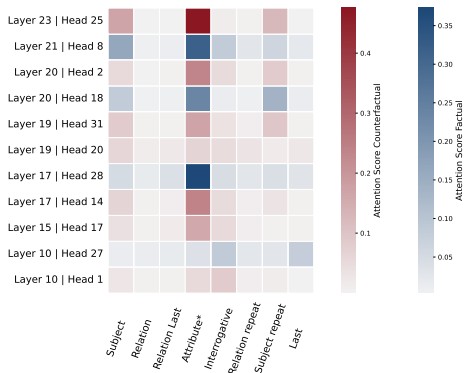
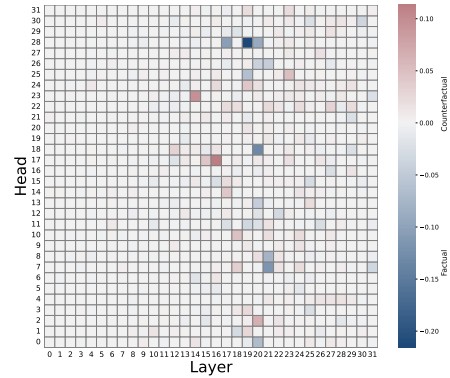

(a) Attention Scores of Key Heads at Last Position for Pythia 6.9B on QnA dataset

(b) Contribution to Δcofa by Pythia 6.9B heads

Figure 21: Logit inspection for Pythia 6.9B, per attention head, on the "QnA" Dataset.

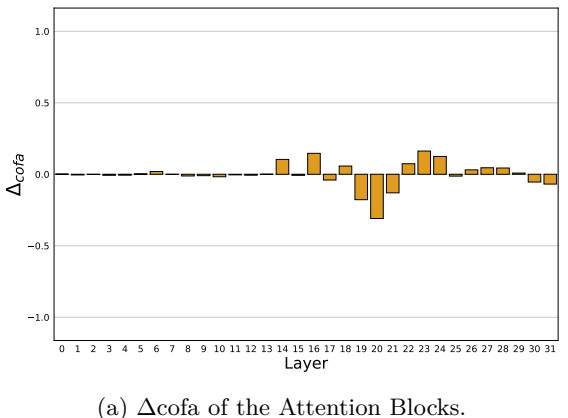
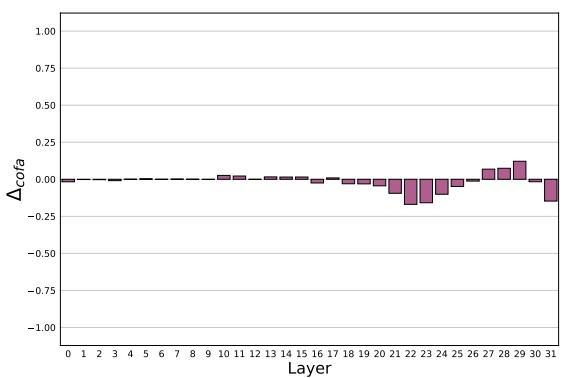

(a) Δcofa of the Attention Blocks.

(b) Δcofa of the MLP Blocks.

Figure 22: Logit inspection of the aggregate impact of attention and MLP blocks on Δcofa for Pythia 6.9B on the "QnA" Dataset.

## B.3 Llama 3.1 8B Plots

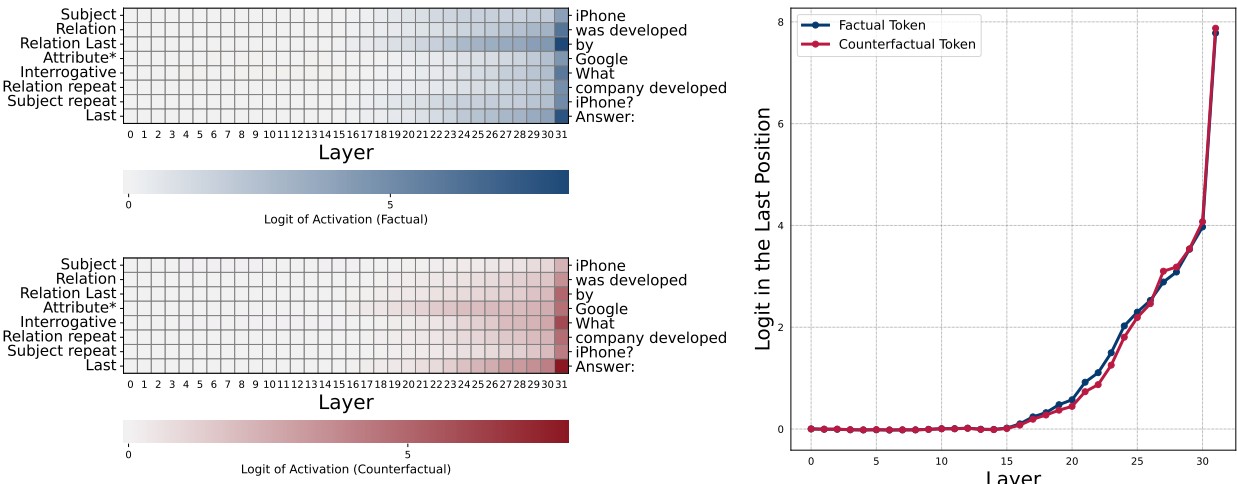

Figure 23: Logit inspection for Llama 3.1 8B on the "QnA" Dataset.

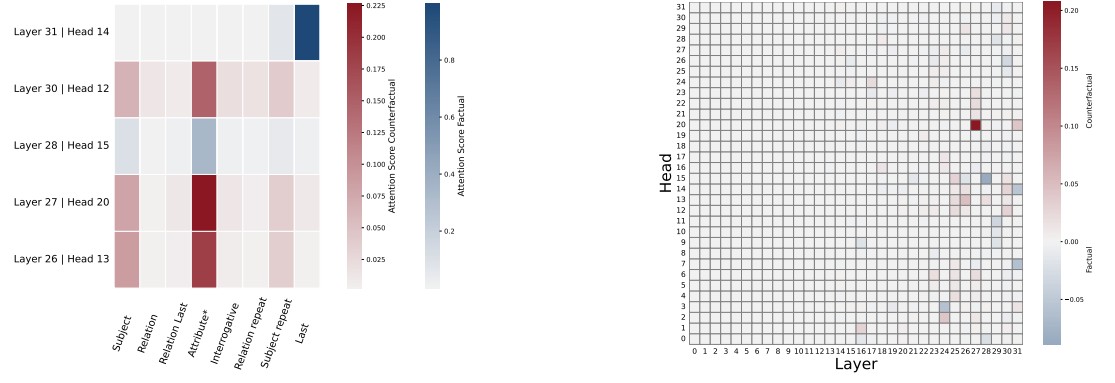

(a) Attention Scores of Key Heads at Last Position

(b) Contribution to Δcofa by Llama 3.1 8B Heads

Figure 24: Logit inspection for Llama 3.1 8B, per attention head, on the "QnA" Dataset.

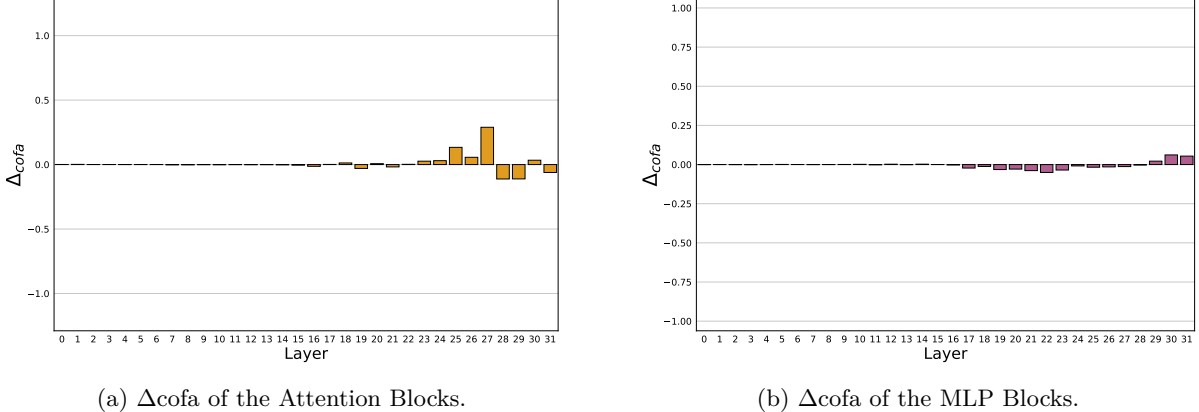

(a) Δcofa of the Attention Blocks.

(b) Δcofa of the MLP Blocks.

Figure 25: Logit inspection of the aggregate impact of attention and MLP blocks on Δcofa for Llama 3.1 8B on the "QnA" Dataset.

