# OpenReview forum: "On the Generalizability of "Competition of Mechanisms: Tracing How Language Models Handle Facts and Counterfactuals""
_TMLR — Accepted by TMLR_

### Review · Reviewer_HR5M · 2025-03-08

**Summary Of Contributions:**

The paper aims to reproduce the study by Ortu et al. 2024. It extends the original study by considering larger LLMs such as Llama 3.1 8B  it, evaluating the impact of the prompt structure, and the relevance of specific domains. The paper is able to reproduce the core claims made by Ortu et al. 2024.

**Audience:**

Yes

**Claims And Evidence:**

Yes

**Requested Changes:**

What I would like to authors to comment on, either in the rebuttal or (if necessary) by revising the paper, is the consequences of their findings. Now, that we learned smth. about the behavior of LLMs, what can we do with these insights (in the real world)? Just some suggestion, feel free to extend/change: Can we use this to improve/"fix" LLMs or prompts? Did we learn smth. about their limitations such that we should not use them in certain cases? etc.

Minor:
Color coding in the layer plots (e.g. Figure 1) could be made more clear. Explicitly stating what the color is encoding (in the legend of the plot, not in the caption of the Figure) could enhance readability even further. I would change "Logit of ?" to "Logit of activation ?", assuming that the activation is color-coded.

**Strengths And Weaknesses:**

Strengths
-------------
The paper is well-written and easy to read even for people who do not have a strong background in this research area. The authors run a sound reproduction study and come up with interesting and convincing extensions that are evaluated as well.


Weaknesses and Recommendations for Improvement
------------------------------------------------------------------------
By nature, the authors focus on "logit inspection" as a tool for analyzing the LLMs. I agree with the authors, that just using "logist inspection" is somewhat limiting and it would be interesting to use more recent methods for analyzing LLMs. I guess this is too much for a revision.

---

> ### Author Response · Authors · 2025-04-12
> **Consequences of our findings**
>
> Thank you for the review!
>
> While our study focused more on the experimental setting laid out by Ortu et al., there are some broader trends we can extract from it. In practice, people rarely try to purposefully influence the output of an LLM to match some notion they have. However, many people might inadvertently pose biased questions to LLMs that would tip them towards answering incorrectly, e.g. by providing opinions or facts supporting their point of view within the prompt. If one treats our experimental setup as a proxy for this phenomenon, our results would suggest that LLMs, especially smaller ones, are rather "impressionable" and readily change their output based on the context, especially when they get to repeat it verbatim. Notably, we saw that premise words like "Fact Check:" and "Review:" are particularly potent at getting GPT-2 small to repeat the incorrect information, even though they provide a rather neutral context. However, we would be reluctant to generalize such a claim to bigger and newer models, since they are often fine-tuned to differentiate between the (biased) context of the user prompt and their own autoregressive output.
>
> With regard to "fixing" LLMs, our domain analysis suggests that this is more difficult than anticipated. While Ortu et al. present mechanism competition as a single copy mechanism versus a single factual recall mechanism, we see that prompts from different domains lead to different attention heads contributing to the factual or the counterfactual token. Furthermore, ablating the two heads that are the biggest contributors on the aggregate leads to varying results when the prompts are grouped by domain topic. Our results thus suggest that the role of each attention head changes depending on the prompt domain, and that LLMs are unlikely to become broadly less biased or inclined to copy from the context by simply ablating a small number of attention heads.
>
> We plan on revising the paper to extend the discussion along the lines of the above comment, as well as to make the figures a bit clearer as you suggested, after all reviews are in.

---

> > ### Comment · Reviewer_HR5M · 2025-04-14
> >
> > Sounds great. Thanks for the clarification.
> >
> > Looking forward to the revision.

---

### Review · Reviewer_zzWp · 2025-04-27

**Summary Of Contributions:**

This is a reproduction of the paper "Competition of Mechanisms: Tracing How Language Models Handle Facts and Counterfactuals." The authors were able to reproduce the results in the original paper by carefully preprocessing their data, and extending the study in three main directions: 1) generalization to larger models, 2) changing the prompt keywords, and 3) varying the prompt domains to check the robustness of the findings.

**Audience:**

Yes

**Claims And Evidence:**

Yes

**Requested Changes:**

The authors should at least clarify 1) above.

**Strengths And Weaknesses:**

The paper performed a sequence of thorough experiments to demonstrate the generalizability of the methodology proposed in the original paper. The experimental design was comprehensive, and the results are interesting to read.

1) Since the original paper is a recent paper, it would be useful for the authors to comment on the importance of reproducing this paper.

2) Additionally, I believe the non-robustness aspects of the original paper shed light on why the methodology might not be a reliable mechanism in evaluating the language model's counterfactual ability. Explicitly commenting on why the method fails (from a methodological perspective) to generalize could significantly strengthen the paper. But it might be beyond the scope of this paper.

---

> ### Author Response · Authors · 2025-05-19
> **Response on work importance and logit lens robustness**
>
> Thank you for the review!
>
> Pragmatically speaking, our reproduction of Ortu et al. was conducted with the goal of submitting it to MLRC (reproml.org). The main criteria for submissions are to reproduce a paper that has been published the year prior as a full paper at a major conference or journal. Since Ortu et al. was published as a full paper at ACL last year, it fit the requirements, which definitely influenced our choice.
>
> Beyond that, we think the field of knowledge conflicts and factual recall in general is very current, considering the relevance to techniques such as retrieval augmented generation (RAG), which have recently become a core part of all modern search engines. Understanding why a model might use information from model memory versus the context and vice versa can thus be crucial for tuning models to faithfully answer queries with a set of given search results.
>
> As such, we think reproducing Ortu et al. was a worthwhile endeavour. Despite a relatively simple setup, the paper presents some interesting results. From a practical point of view, the implied promise is that one can take a base LLM and, using relatively little data, identify a small number of specialized attention heads that control whether the model chooses to rely on in-context information versus information from its pretraining. Furthermore, one can (purportedly) simply scale the output of these attention heads to receive over 12 times more "factual" responses, which, as mentioned above, holds immense potential for usage in RAG, or as a way to make models more resilient to prompt influences.
>
> To be clear, our paper suggests that this promise does not hold. The methodology proposed in Ortu et al. seems to be very sensitive to the particular prompt structure, does not generalize meaningfully across datasets of different domains and, at least when using only logit lens, fails to identify specialized attention heads in newer models such as Llama 3.1 8B. While we refrain from claiming that our findings completely disprove the effectiveness of the methodology (e.g. broadly speaking, Llama 3.1 8B does seem to exhibit a competition between the factual and counterfactual mechanism), it appears to lack the necessary robustness for the wide applicability we had hoped for. Further research is required to translate the findings of Ortu et al. to a more practical setting.
>
> Overall, we agree that a comment on the broader impact of our work beyond "Ortu et al. generalizes/does not generalize" would be meaningful to our audience. As such, we have added subsection 5.3 ("Practical considerations") to our discussion, which is inspired by our response to your and your fellow reviewers' questions.
>
> On your second point, it is fair to question the robustness of logit lens analysis, especially as model architectures keep growing larger and more complex compared to the GPT models logit lens was originally proposed for. Unfortunately, given that the method is built mostly on top of empirical evaluations rather than a strong theoretical framework, our results can hardly tell us why the patterns from GPT-2 do not seem to generalize. Considering the spike of logit activations for both the factual and counterfactual token in the last layer of Pythia 6.9B and Llama 3.1 8B, one could guess that these larger models learn to perform some sort of representational shift just before their final output. This could be the case if a different internal vector basis allows the models to more effectively group concepts than the one provided by the embedding matrix. However, it could also be the case that the main assumption of logit lens - that the residual stream approximates the final distribution long before the end of the network - only holds for a small number of layers before the output, and that larger models simply have more layers to utilize before that happens. To avoid speculation, we leave this topic for future work.

---

> > ### Comment · Reviewer_zzWp · 2025-05-20
> > **revision**
> >
> > Thanks for the added paragraph.

---

### Review · Reviewer_jzkd · 2025-05-06

**Summary Of Contributions:**

- This LLM interpretability paper builds upon the framework introduced by Ortu et al., focusing on how large language models (LLMs) internally manage two competing processes:
	- Factual Knowledge Recall, where the model retrieves stored factual information.
	- In-Context Adaptation to Counterfactual Statements, where the model adapts dynamically to explicitly provided counterfactual information.
- For simplicity, these are referred to as "factual" and "counterfactual" mechanisms.
- The authors successfully replicate most of Ortu et al.’s original findings using GPT-2 and Pythia models, confirming the specific locations and roles of model components involved in these mechanisms.
- They further test whether the original conclusions hold when applied to the larger LLaMA-8B model, demonstrating that the identified patterns remain consistent even in more advanced models.
- Additionally, the authors investigate how variations in prompt structure—such as altering key words in the prompts or omitting the repeated counterfactual statements—impact the strength and interaction of these competing mechanisms.
- The authors also assess whether these insights are consistent across different domains, highlighting the broader applicability of the interpretability methods from Ortu et al.

By confirming previous results and expanding the analysis to additional models, prompt variations, and diverse domains, this paper deepens our understanding of how LLMs handle conflicting factual and counterfactual information internally.

**Audience:**

Yes

**Claims And Evidence:**

Yes

**Requested Changes:**

- Clarification for the 3rd point above.
- Discussion on limitations of Logit Lens. Possibly incorporate or suggest additional interpretability methods (e.g., activation patching, probing classifiers) to complement your analysis.

**Strengths And Weaknesses:**

### Strengths
- The authors rigorously reproduce and validate or invalidate earlier findings from Ortu et al., extending analyses effectively to larger models (LLaMA-8B), different prompt configurations, and multiple domains. This thorough investigation would be useful to the community and provides a thorough resource for the study of the competition-of-mechanisms framework.
- The detailed exploration of how subtle prompt changes influence internal mechanism dynamics is valuable, offering practical insights for future interpretability research and helping guide better prompt design.
- The paper's systematic cross-domain analysis effectively reveals how factual information subtly "leaks" into model predictions despite clear counterfactual contexts.
### Weakness
- The study closely follows the interpretability framework established by Ortu et al., with minimal methodological advancements. Introducing new analytical techniques or perspectives could provide deeper insights and strengthen the contribution to the field.
- It is somewhat difficult to clearly determine if some of the inconclusive findings stem from limitations of the logit lens technique itself or if they reflect inherent complexities in identifying these mechanisms. Integrating other interpretability methods like activation patching or probing classifiers could help clarify the results and strengthen the overall claims.
- Page 6: "However, the claim that factual information is stored in the subject position does not seem to be backed up by our results or those of the original paper. Instead, the positions most strongly influencing the factual prediction seem to be the last token of the first relation and the tokens of the first relation themselves." -> this makes sense looking at Figure 1. However, it also holds for Figure 6 in the Appendix so I am not clear on the following sentence "for GPT-2 small, factual information is predominantly encoded in the subject position during early layers, while counterfactual information is encoded in the attribute position". The subject position has a lighter blue color in both figures. Additional clarification needed.

---

> ### Author Response · Authors · 2025-05-16
> **Clarification and response**
>
> Thank you for the review!
>
> We wholeheartedly agree that the results we obtained using logit lens analysis are inconclusive and a bit unsatisfactory for models larger than GPT-2 small. In fact, one of the goals of the study was to investigate whether the results from Ortu et al. generalize well to newer, larger transformers. Whether the discrepancies in our findings are due to the architecture and scale of the models, or whether they stem from logit lens being an unreliable interpretability method, is an incredibly interesting and important question. Unfortunately, we feel it is outside the scope of our paper. We acknowledge this limitation in our discussion and we point towards avenues for future work, such as applying a modified variant of logit lens (like tuned lens), or by approaching the problem of competing mechanisms from a different angle, such as with circuit analysis.
>
> On your point about the subject position, we agree that the language we used is confusing. Ortu et al. found that, in layers near the end (output) of the network, the last tokens before each attribute are the primary contributors to both the factual and the counterfactual logit. However, they claim that in early layers (near the input) the factual logit is mostly encoded in the subject position, while the counterfactual logit is mostly encoded in the attribute position. We concur with Ortu et al. on GPT-2; in the first 4-5 layers of our Figure 6 in the Appendix, the tokens in the subject position are responsible for slightly higher factual logit values than the other positions (though the difference is hard to spot), and the tokens in the attribute position are responsible for the counterfactual logit values. Meanwhile, in the last few layers, the information for both tokens is mostly encoded in the "Relation Last" and the "Last" positions. However, for Pythia all positions in the early layers seem to be contributing equally little, which is a piece of analysis missing from the original paper and points towards the results not being very generalizable to deeper models.
>
> We thank you for pointing out the ambiguity in our description. We have revised it so the first paragraph in the "Positional Information Encoding" subsection (bottom of page 6) now reads as follows:
>
> "Our results confirm that, in the early layers of GPT-2 small, factual information is spread out but most strongly encoded in the subject position, while counterfactual information is predominantly encoded in the attribute position (Figure 6 in the Appendix). As in the original study, in later layers both types of information become strongly concentrated in the last token position, with logit values showing a monotonic increase across layers."
>
> Let us know if that clarifies it sufficiently.

---

> ### Comment · Reviewer_jzkd · 2025-05-26
> **Reply to Clarification**
>
> Thanks for the clarification and response.
>
> - The limitation for logit lens isn't emphasized enough in the Limitations and Future Work section. The only sentence I see is with respect to the Llama 8B model. I think the paper can be benefit from calling out the limitations of logit lens better.
>
> - Appreciate the rewording on Page 6 but I am still confused by the figures. Let's look at the color (darker color represents higher logit values). The token "by" has the darkest color for the later layers which is neither the subject or attribute position . Can you please explain how this sentence is true "spread out but most stronly encoded in the subject position" ?
> Am I missing something here?
>
> - Also were the plots regenerated because the original colors and the new colors are slighly different https://draftable.com/compare/ywkqhTaznWUX

---

> > ### Author Response · Authors · 2025-05-28
> > **Clarification on plots and limitations**
> >
> > Hello again,
> >
> > The plots were indeed regenerated, as we stated in our [revision comment](https://openreview.net/forum?id=15keyzQj9h&noteId=BEvPmQiLHn). We used the same data as before, but changed the colour palette to be darker, so we can better show the tiny imbalances in logit levels we are interested in.
> >
> > Regarding your second question about the figures, let us look at the plot for factual logit activations by position for GPT-2 small. You can refer to the [following visual](https://imgur.com/a/MrpjZ6a) for guidance.
> >
> > On the bottom of page 6, we state the following: "Our results confirm that, *in the early layers* [emphasis mine] of GPT-2 small, factual information is spread out but most strongly encoded in the subject position...". Looking at the plot, ignoring the latter half of it and only focusing on the first few layers, we can see that all of the cells are relatively near in colour. However, the cells for the subject position are one shade darker than the others, as their value is just a bit higher. Continuing with our position: "As in the original study, *in later layers* both types of information become strongly concentrated *in the last token position* [emphasis mine]". Looking at the plot again, this is a lot clearer: in the last couple of layers, while all cells become darker, the ones in the "last" and "relation last" positions become significantly darker, approaching dark blue with our new colour scheme.
> >
> > We agree that the intensity of these two phenomena is very different, with the activations in the later layers being a lot stronger than the ones early on. However, we wanted to extend the most charitable possible interpretation of Ortu et al., and so their claims broadly hold on this matter. Furthermore, if we compare our plot with their plot, it is almost identical (at least before the colour change), so we are confident in confirming their data. We do not think this is misleading for two reasons. First, we include all the plots we generated, so our readers can visually inspect them and draw their own conclusions. Second, in our domain analysis in section 4.5, it becomes clear that the activation at the "subject" position is dependent on the domain of the questions, and we believe it to be a fault with the dataset. In Figure 3, one can more clearly see the importance of the "subject" token for questions in the domain "Autos and Vehicles", compared to "Arts and Entertainment". In the end, our additional findings disprove the idea that the "subject" token is important for the competition of mechanisms more broadly, and we hope that that is the takeaway message.
> >
> > On the limitations of logit lens analysis, the reason we do not explicitly claim something along the lines of "logit lens is an insufficient method to properly evaluate the competition of mechanisms" is because our results do not prove or disprove this. The reason we talk about Llama specifically is because for GPT-2 and Pythia, logit lens *does* find specialized attention heads that severely impact the output if ablated (as much as that is not very generalizable or robust across questions, prompts or architectures). In Llama 3.1 8B, logit lens does not find specialized attention heads, but without further analysis it is impossible to say if this is a limitation of logit lens, Llama or something else entirely, like the prompt structure or the questions themselves. However, if you have strong grounds to believe that this is an inherent problem specifically with logit lens based on our results, we would love to hear them and possibly amend our discussion.
> >
> > Let us know if you have any further concerns.

---

> > > ### Comment · Reviewer_jzkd · 2025-06-02
> > > **Thanks for the Clarification**
> > >
> > > - Thanks for the clarification. It clarifies my misinterpretation.
> > > (Nit: It would help a potential reader if you could mark in the diagrams the way you marked in the visual here, the distinction between early and later layers. )
> > >
> > > - I am fine with not making explicit claim about logit lens.

---

### Author Response · Authors · 2025-05-17
**Revision**

Dear reviewers,

We have submitted a revision to our work. We have added a new subsection (5.3) in the discussion addressing the wider impacts and takeaways from our work. We have also reworded the second paragraph of section 4.1 for clarity. Finally, we have made the charts a bit darker to make them more readable when comparing low values for logit activations.

Part of the new section is inspired by the comments from reviewer zzWp, a response to which will be published on Monday.

Thank you for your patience.

---

> ### Comment · Reviewer_HR5M · 2025-05-20
>
> Thanks. This resolves all my questions and concerns.
>
> ### Minor request regarding the references
>
> > Thomas Wolf. Transformers: State-of-the-art natural language processing. arXiv preprint arXiv:1910.03771, 2020.
>
> Please use the peer-reviewed reference instead of the arXiv preprint:
> ```
> Wolf, Thomas, et al. "Transformers: State-of-the-art natural language processing." Proceedings of the 2020 conference on empirical methods in natural language processing: system demonstrations. 2020.
> ```

---

> > ### Author Response · Authors · 2025-05-20
> > **Reference update**
> >
> > Thank you for pointing out the faulty reference. We have updated our references to ensure we correctly represent papers that have been accepted at appropriate venues, specifically Wolf et al. ("Transformers: State-of-the-art natural language processing") and Zhang et al. (Mquake: Assessing knowledge editing in language models via multi-hop questions).

---

> > > ### Comment · Reviewer_HR5M · 2025-05-20
> > >
> > > Thanks!

---

### Decision · Action_Editor_pnKL · 2025-06-09

**Recommendation:** Accept as is

**Audience:**

Yes

**Audience Explanation:**

Readers interested in causality and large language models will appreciate this replication study.

**Claims And Evidence:**

Yes

**Claims Explanation:**

This is a replication study of the work by Ortu et al. (2024), and it extends Ortu et al. (2024) in different directions demonstrating the generalizability of the results.